# Nitrogen-Doped Porous MXene (Ti_3_C_2_) for Flexible Supercapacitors with Enhanced Storage Performance

**DOI:** 10.3390/molecules27154890

**Published:** 2022-07-30

**Authors:** Xin Tao, Linlin Zhang, Xuedong He, Lingzi Fang, Hongyan Wang, Li Zhang, Lianghao Yu, Guang Zhu

**Affiliations:** 1School of Mechanics and Optoelectronic Physics, Anhui University of Science and Technology, Huainan 232001, China; taox5802311013@163.com (X.T.); 17805572596@163.com (L.Z.); 2Key Laboratory of Spin Electron and Nanomaterials of Anhui Higher Education Institutes, School of Mechanical and Electronic Engineering, Suzhou University, Suzhou 234000, China; fanglz0@163.com (L.F.); suzhouwhy@163.com (H.W.); zhlisuzh@163.com (L.Z.); 3Key Laboratory of Leather of Zhejiang Province, College of Chemistry and Materials Engineering, Wenzhou University, Wenzhou 325035, China; xuedonghe7878@gmail.com

**Keywords:** flexible supercapacitors, nitrogen-doped 3D MXene, electrospinning

## Abstract

Flexible supercapacitors (FSCs) are limited in flexible electronics applications due to their low energy density. Therefore, developing electrode materials with high energy density, high electrochemical activity, and remarkable flexibility is challenging. Herein, we designed nitrogen-doped porous MXene (N-MXene), using melamine-formaldehyde (MF) microspheres as a template and nitrogen source. We combined it with an electrospinning process to produce a highly flexible nitrogen-doped porous MXene nanofiber (N-MXene-F) as a self-supporting electrode material and assembled it into a symmetrical supercapacitor (SSC). On the one hand, the interconnected mesh structure allows the electrolyte to penetrate the porous network to fully infiltrate the material surface, shortening the ion transport channels; on the other hand, the uniform nitrogen doping enhances the pseudocapacitive performance. As a result, the as-assembled SSC exhibited excellent electrochemical performance and excellent long-term durability, achieving an energy density of 12.78 Wh kg^−1^ at a power density of 1080 W kg^−1^, with long-term cycling stability up to 5000 cycles. This work demonstrates the impact of structural design and atomic doping on the electrochemical performance of MXene and opens up an exciting possibility for the fabrication of highly FSCs.

## 1. Introduction

Recently, the dimensions of electronics have changed from their inherent rigid and rectangular shapes to more intelligent form factors such as flexible electronic products, smart bracelets, artificial skins, and wearable sensors [1,2,3,4]. These intelligent next-generation electronic products have become a hot topic due to being lightweight and conforming to the curves of the body [5]. Among the numerous components of wearable devices, the real bottleneck hindering flexible electronics from becoming ubiquitous in practical products is the construction of flexible and deformable energy-storage devices. This requirement, in turn, promotes intensive research on designing and fabricating highly flexible energy-storage devices such as batteries and supercapacitors [6,7,8]. For this reason, various flexible energy-storage devices such as FSCs are being developed.

Among the fabrications of FSCs’ devices, preparing electrode materials with high flexibility has proven to be one of the most critical challenge [9], which is vital for FSCs without binder and conductive additives. Binder and conductive additives inevitably cause some agglomeration and significantly reduce the volumetric and gravimetric capacitance [10]. Among the various techniques for the design of FSCs without binder and conductive additives, electrospinning nanofibers usually possess unique properties such as high specific surface area, interconnected microfiber structure, high tortuosity, high permeability, and lightweight materials, which are well-suited for the design of flexible electrodes. Additionally, with unique structural and ion-transport characteristics, one-dimensional nanomaterials have attracted much interest in flexible electrodes. To date, a variety of materials have been successfully demonstrated to fabricate flexible electrodes by electrospinning [11,12,13]. Carbon materials, such as graphene (G) [14], activated carbon (AC) [15], and carbon nanotubes (CNTs) [16], have excellent electrical conductivity but limited mass capacitance due to the electrochemical double layer capacitance (EDLC). Transition metal oxides (TMOs) such as MoO_3_ [17], MnO_x_ [18], V_2_O_5_ [19], and RuO_2_ [20] usually possess incredibly high theoretical specific capacitances. However, sluggish kinetics and poor electrical conductivity lead to severe capacitance fading and jeopardize the commercialization of FSCs. The metal-organic framework (MOF) usually has the advantages of a large specific surface area, diverse structure, and adjustable pore size. However, MOF materials also suffer from poor electrical conductivity, which in turn reduces energy density [21,22,23,24]. Therefore, finding a suitable material with high electrochemical performance accompanied by a flexible electrode is a crucial step in promoting the development of FSCs.

As a new member of the 2D-materials family, transition metal carbides, nitrides, and carbonitride, also known as MXene, have attracted widespread attention due to their outstanding electrochemical, optical, mechanical, and electronic properties, thus showing huge potential in energy storage, sensors, and catalysis [25,26,27]. Nevertheless, the van der Waals interactions between MXene flakes, thus, result in restacking [24]. It suddenly curtails the ion transport among electrodes and severely restricts the full advantage of the surface-active group [28]. Macro-assembly is important for translating individual MXene nanosheets into practical macroscopic-material properties. Recent work has shown that template-assisted synthesis of porous MXene framework has emerged as a valuable solution to tackle aggregation [29,30]. Benefiting from the electrostatic force between opposite charges (positively charged templating agents and negatively charged MXene flakes). Sun and co-workers designed a versatile nitrogen-doped MXene using positively charged melamine as the template and nitrogen source for application in direct ink writing [31]. Besides, investigations on developing well-dispersed MXene nanosheets to construct FSCs, by electrospinning targeting high volumetric capacitances and flexible characteristics, are still in their infancy.

Herein, we report on the direct electrospinning of FSC featuring high volumetric energy densities based on crumpled N-MXene nanosheets. The designed, which crumpled N-MXene with uniform nitrogen inclusion and rich mesopores structure, can be attributed to a facile templating route. Nitrogen atoms are beneficial to improving their redox reactivity and the electron transport of MXene, thereby enhancing its storage properties according to rate capability and cyclic stability. Moreover, the 3D porous system integrated a high surface area, thus exposing more chemically active sites. As a result, the as-assembled SSC exhibited a superior energy density of 12.78 Wh kg^−1^ at a power density of 1080 W kg^−1^. Our work demonstrates the potential of electrospinning technology for the customized design of MXene-based high-performance devices for wearable energy-storage-related fields.

## 2. Experimental Section

### 2.1. Reagent and Materials

Ti_3_AlC_2_ powder was purchased from 11 Technology Co., Ltd. (Changchun, China). The lithium fluoride (LiF) was purchased from Aladdin Shanghai Co., Ltd. (Shanghai, China). Concentrated hydrochloric acid (HCl) was purchased from Shanghai Zhenqi Chemical Reagent Co., Ltd. (Shanghai, China). The polyacrylonitrile (PAN), melamine, *N*-dimethylformamide (DMF), formic acid, and formaldehyde solutions were purchased from Shanghai Maclean Biochemical Technology Co., Ltd. (Shanghai, China). All reagents (analytical grade, AR) were directly used without further purification.

### 2.2. Synthesis of MF Sphere

First, 10 mL of formaldehyde and 200 mL of deionized water were added to the round bottom flask and heated at 80 °C. After the temperature stabilized, 2.5 g of melamine was added to the solution under constant stirring until completely dissolved. After 20 min of pre-polymerization, 0.25 mL formic acid was slowly added dropwise to the above solution to adjust the pH value. The liquid rapidly turned milky white and stirred continuously for 40 min to form MF microspheres. The resulting samples were collected and washed several times by centrifugation with distilled water and anhydrous ethanol and then dried in a vacuum oven at 60 °C for 24 h. Finally, the collected white solid powder was about 0.632 g.

### 2.3. Synthesis of N-MXene and N-MXene-F

MXene dispersions use the preparation process of our previous work [31]. To prepare MXene@MF nanocomposite, specifically, 40 mL (2.5 mg mL^−1^) of MXene dispersions was ultrasonically dispersed in 100 mL of deionized water. Then, the MXene solution was slowly added to 100 mL of MF dispersion (8 mg mL^−1^). After sufficiently stirring, MXene@MF nanocomposites were obtained by repeated centrifugal washing and freeze-drying. Then, it was annealed at 550 °C with a ramp of 5 °C min^−1^ under N_2_ atmosphere for 2 h to obtain N-MXene. The collected black powder was about 0.14 g. For the preparation of N-MXene-F. Firstly, 0.7 g MXene@MF and 9 g DMF were sonicated for 30 min, and 0.8 g polyacrylonitrile was added and stirred for 12 h at 50 °C. Subsequently, the obtained mixture was added to the injection for the spinning process to get MXene@MF-F. Finally, the product was annealed at 650 °C for 3 h with a ramp of 5 °C min^−1^ to acquire N-MXene-F. For comparison, MXene-F were prepared in the same way, except that MF was not added.

### 2.4. Characterization

The morphological characteristics of all the prepared samples were tested by scanning electron microscopy (SEM, Nova Nanosem 200 system, 10 kV, FEI, Hillsboro, OR, USA). In the SEM test, the sample is applied directly to the conductive tape without any other treatment. Transmission electron microscopy (TEM, JEOL-2100 with an accelerating voltage of 200 keV, Jieou Road (Beijing) Technology and Trade Co., Ltd., Beijing, China) examination of the surface morphology. Energy dispersive X-ray spectroscopy (EDX) by high-resolution TEM (HRTEM, Jieou Road (Beijing) Technology and Trade Co., Ltd., Beijing, China) to assess the presence of elemental composition. X-ray diffraction (XRD, Bruker D8 Advance with Cu Kα radiation source, Karlsruhe, Germany) for the detection of the composition and crystal structure of the sample, set at an angle range of 5° to 80° and a scan rate of 6° min^−1^. Monochromatic X-ray photoelectron spectroscopy (XPS, Kratos Axis Ultra DLD, Kratos, Manchester, UK) using 22.4 W Al kα radiation (1486.7 eV) to determine the surface composition. XPS data were processed in XPS Peak software with Shirley as the background and C1s as the reference for charge compensation. The nitrogen adsorption/desorption isotherm (Micrometrics ASAP 2020, Micromeritics Instrument Corporation, Norcross, GA, USA) was measured by a physical adsorption meter. The specific surface area was analyzed by the Brunauer–Emmett–Teller (BET) method, and the pore distribution was calculated by the density flooding theory (DFT) method. The BET-specific surface area was calculated in the range of P/P_0_ of 0.05–0.3.

## 3. Results and Discussion

The main fabrication process of the highly flexible electrode material is illustrated in Figure 1. As shown on the left side of the picture, firstly, during the whole wet-stripping (HCl + LiF) process, a few layers of MXene flakes were obtained, and a significant Tindal effect proved the formation of colloidal solution (Appendix A) [32]. Subsequently, the prepared MF sphere was added to the MXene suspension, and the negatively charged MXene flakes assembled rapidly and spontaneously with the positively charged MF to form MXene@MF precipitates (Appendix A). Then, the freeze-dried MXene@MF powder was annealed to remove the MF to obtain three-dimensional porous nitrogen-doped MXene. MF not only provided a template for the pore-making design hindering the spontaneous stacking of MXene but also defined the rich doping of MXene with nitrogen atoms [33]. Finally, the highly flexible N-Mxene-F nanofiber film is obtained by the direct electrospinning of MXene@MF and PAN as raw materials and further annealing of the resulting product.

The SEM image in Figure 2a shows that MXene@MF has a uniformly wrapped structure, and the 2D MXene nanosheets form a wrinkled and curved morphology due to the presence of MF. The curved structure of this MXene flake was not destroyed after the template removal process, forming the crumpled 3D porous MXene, as shown in Figure 2b. This porous honeycomb structure effectively inhibits the self-stacking of MXene sheets while promoting complete electrolyte penetration. The SEM images show, in Figure 2c, that N-MXene-F exhibits nanofiber bead-chain-like morphology. In addition, the growth orientation of all nanofibers can be seen in Appendix A as random, while we also checked the size of nanofibers at 67 locations, and the results are shown in Appendix A. The results show a distribution ratio of 62.5% for the size of nanofibers between 200–300 nm and an average diameter of about 230 nm in the counted range.

This electrospun sample can produce free-standing films. It can be seen in Appendix A that the MXene@MF-F films can maintain their mechanical integrity after annealing. It can be easily folded into various shapes, indicating that as-prepared N-MXene-F films have potential applications in flexible electronics.

As shown in Figure 2d, the microstructure of the as-prepared N-MXene was analyzed using high-magnification TEM images, from which the satisfactory formation of the porous structure could be further demonstrated. Figure 2e shows bright lattice fringes with a lattice spacing of 0.35 nm shown in the HRTEM image, corresponding to the interlayer spacing. To further determine the atomic doping and elemental distribution of N-MXene, we performed a mapping probe by EDX analysis in scanning transmission electron microscopy (STEM) mode. As shown in Figure 2f, the N, O, and Ti elements are uniformly distributed, strongly demonstrating that nitrogen atoms may be successfully doped into the MXene framework.

The X-ray diffraction (XRD) patterns of MXene and N-MXene are shown in Figure 3a. Both N-MXene and MXene show distinct characteristic diffraction (002) peaks. Low-angle range XRD patterns of MAX, etched MXene, MXene, and N-MXene show a shift in the (002) peak (Appendix A). It is easy to observe that some diffraction peaks of etched MXene, MXene, and N-MXene are weakened or disappeared, compared to the XRD patterns of the MAX, indicating that the etching and sonication processes have destroyed the multilayer structure of MXene [31]. In particular, the (002) peak of the MAX phase is around 9.45°, while the (002) peaks of etched MXene, exfoliated MXene, and N-MXene are around 7.22°, 6.41°, and 5.98°, respectively. The significant shift of the (002) peak to a small angle indicates that the MXene layer spacing has become broader. To determine the specific surface area and pore structure of the material, the specific surface area of the samples was analyzed by nitrogen adsorption and desorption measurements, as shown in Figure 3b, and the pore size distribution is determined in Figure 3c. MXene, N-MXene, and N-MXene-F all exhibit type IV isotherms. At a relative pressure of 0.4 to 0.8, both N-MXene and N-MXene-F exhibit a hysteresis loop, indicating the presence of mesopores, while MXene has no significant hysteresis loop, indicating the lack of mesoporous structures, which is confirmed in Figure 3c [34]. All samples exhibit a significant hysteresis loop attributed to macropores at relative pressures from 0.8 to 1.0, which is attributed to macropore formation. Due to the tight packing of MXene-layered nanosheets, resulting in its lower BET specific surface area (3 m^2^ g^−1^), N-MXene with porous structure fabricated from MF template possesses more large BET specific surface area (185 m^2^ g^−1^). In addition, the fiber-bead-chain N-MXene-F, prepared by electrospinning, possesses an exciting BET-specific surface area (197 m^2^ g^−1^). The rich mesoporous structure and the increased specific surface area allow N-MXene and N-MXene-F to have more ion-accessible regions during electrochemical processes, which can enhance electrochemical energy storage capacitance [31] and, further, collected XPS patterns to analyze the chemical composition of N-MXene. The XPS results in Figure 3d confirm the presence of C, N, Ti, and O. Meanwhile, the atomic concentrations detected, as shown in Appendix A, are 56.97%, 10.37%, 12.21%, and 20.44%, respectively. The high-resolution Ti 2p spectrum shows deconvoluted peaks at 457.8, 455.8, and 454.6 eV in Figure 3e, assigned to Ti-O, Ti-N, and Ti-C bonding, respectively. Figure 3f shows the N 1s spectrum and the colocalization of pyrrole N (400.3 eV), pyridine N (398.5 eV), and Ti-N (397.9 eV) can be remarked [31]. Notably, the presence of C-N bonds was found in the high-resolution C 1s spectrum (Appendix A), and the presence of N-O bonds was found in the high-resolution O 1s spectrum, further demonstrating the successful doping of nitrogen atoms.

To further understand the effect of 3D porous structure and nitrogen atom doping on the electrochemical behavior of MXene. Cyclic voltammetry (CV), galvanostatic charge–discharge (GCD), and electrochemical impedance spectroscopy (EIS) tests were carried out to evaluate the electrochemical performance of N-MXene and MXene in a three-electrode system. The CV curves of N-MXene and MXene are shown in Figure 4a. The results show that the N-MXene curve exhibits a larger area and higher response current than the pristine MXene, indicating a higher charge-storage behavior and faster kinetics. In addition, the CV performance of N-MXene at different scan rates (1–100 mV s^−1^) was also tested. Figure 4b shows that the CV curves of the as-prepared N-MXene electrodes exhibit leaf-like shapes at different scan rates, indicating that the capacitance is ascribed to the combined contributions of the surface Faraday pseudocapacitance and electric double-layer capacitance. Furthermore, the excellent capacitive properties are maintained, even when the scan rate increases from 1 mV s^−1^ to 100 mV s^−1^, reflecting the excellent capacitive behavior and high rate capability of N-MXene. To make it easier to understand the electrochemical performance of MXene and N-MXene electrodes, Figure 4c illustrates the specific capacitance of N-MXene and MXene electrodes at scan rates ranging from 1 to 100 mV s^−1^. The N-MXene electrode provides a specific capacitance of 292 F g^−1^ at a scan rate of 1 mV s^−1^, while the MXene electrode has only 95 F g^−1^ at a scan rate of 1 mV s^−1^. This means that the specific capacitance of N-MXene is about three times higher than MXene. In addition, the N-MXene electrode exhibits a capacitance of 105 F g^−1^, even at a scan rate of 100 mV s^−1^. We also tested the galvanostatic chargedischarge capability of N-MXene with MXene (Appendix A). The results show that the specific capacitance of N-MXene is 2.5 times higher than that of MXene. Specifically, at a current density of 2 A g^−1^, N-MXene is 176 F g^−1^, and MXene is only 71 F g^−1^. This result is consistent with that of cyclic voltammetry. EIS is a widely used method to estimate the electrical conductivity and energy storage properties of fabricated electrodes. In a typical Nyquist plot, the small semicircle diameter at high frequencies corresponds to the charge transfer resistance (Rct). Figure 4d shows that N-MXene has a smaller semicircular diameter in the high-frequency region, indicating a lesser Rct. After measurement, Rct = 0.13 Ω for N-MXene, and Rct = 0.18 Ω for MXene. In addition, in the low-frequency region of the Nyquist curve, N-MXene exhibits a greater slope, indicating an immense contribution from the diffusion-controlled part and a faster rate of ion entry into the interlayer of the active material. The excellent electrochemical performance of N-MXene materials could be attributed to the following factors: firstly, the crumpled porous 3D structure increases the interfacial area between MXene and electrolyte, exposing more electroactive sites and ensuring high electrochemical utilization of the active material. Secondly, due to the introduction of N dopants in the annealing process (Figure 4e), nitrogen dopants will replace some of the C atoms and end functional groups in the MXene backbone, which can significantly improve the pseudocapacitive contribution. Finally, the synergistic effect of the increased specific surface area and uniform atomic doping leads to excellent electrochemical storage results. To verify the stability of the three-dimensional porous structure. The SEM of the N-MXene electrode after 5000 cycles in the three-electrode system was examined, as shown in Appendix A. The results showed that the structure did not change significantly, indicating that the structure of N-MXene has excellent stability.

The N-MXene-based symmetrical supercapacitor provides further validation of the ideal behavior of the capacitor. Figure 5a illustrates the CV curves of the assembled SSC. An excellent rectangular shape can be seen at various scan rates, indicating excellent capacitance characteristics. In addition, the nearly linear and symmetric GCD curves between 0 and 0.6 V (Appendix A) indicate the excellent coulombic efficiency and outstanding electrochemical reversibility of our assembled SSC device. We calculated specific capacitance and specific areal capacitance at different scan rates, in terms of the area and mass of the total active material of the two electrodes, which are shown in Appendix A. In detail, the constructed SSC exhibits an excellent capacitance of 59 F cm^−2^ area capacitance and 79 F g^−1^ mass-specific capacitance, at a scan rate of 1 mV s^−1^, and still maintains 51 F g^−1^ (66% retention) even when the scan rate increases to 100 mV s^−1^. In addition, the electrochemical performance of MXene-F-based SSC and N-MXene-F-based SSC was also compared (Appendix A). The CV curves of the N-Mxene-F-based SSC contain a larger area at a scan rate of 5 mV s^−1^, and a more extended charge/discharge capability at a current density of 1 A g^−1^ indicates superior performance. The life cycle is an important criterion for assessing the practical application of the electrodes. Figure 5b shows the cycling stability of the N-MXene-F-based SSC. After 5000 cycles at a current density of 10 A g^−1^, capacitance retention of 83% can be easily achieved. EIS tests were performed before and after SSC cycling (Appendix A). It can be seen that the charge transfer resistance after cycling is significantly reduced, which may be due to the sufficient infiltration of the electrolyte. In addition, the fabricated N-Mxene-F-based SSC device achieves excellent energy density. Excitingly, the N-Mxene-F//N-Mxene-F SSC devices can easily achieve a maximum energy density of 12.78 Wh kg^−1^ when the power density is 1080 W kg^−1^, as shown in Figure 5c. More importantly, the energy density can still be maintained at 8.46 Wh kg^−1^ when the power density increases to 10.87 kW kg^−1^. It is worth noting that both the energy density and power density of our SSC devices are higher than those of previously reported SSC devices based on graphene- and MXene-based materials, such as PANI-RGO-ZnO//PANI-RGO-ZnO (5.61 Wh kg^−1^, 403 W kg^−1^) [35], LT-Ti_3_C_2_T_x_//LT-Ti_3_C_2_T_x_ (5.67 Wh kg^−1^, 589 W kg^−1^) [36], MXene/N-CuMe_2_Pc//MXene/N-CuMe_2_Pc (8.84 Wh kg^−1^, 112,3 W kg^−1^) [37], MXene/NCF//MXene/NCF (8.75 Wh kg^−1^, 1871 W kg^−1^) [38], G/MXene//G/MXene (5.7 Wh kg^−1^, 5000 W kg^−1^) [39], and G/Ni//G/Ni (3.03 Wh kg^−1^, 562.5 W kg^−1^) [40]. To further demonstrate the practical application potential of our assembled SSC. We produced flexible SSC devices based on N-MXene-F electrodes for practical applications as proof of principle. As shown in Figure 5d, they consist of two N-MXene-F self-supporting films of the same quality as electrodes; glass fibers infiltrated with 2M H_2_SO_4_ are used as separators and electrolytes; and PI films encapsulate the entire device. To further evaluate the flexibility of N-MXene SSC, electrochemical performance tests were conducted at different bending degrees. As shown in Figure 5e, the bending angles were set to 0°, 30°, 60°, and 90° and tested for CV performance. It was found that the CV curves did not change significantly (Figure 5f). To further evaluate the flexibility performance of N-Mxene-F-based SSC devices, we carried out 12 tests from 0° to 90° to 0° and tested the EIS at 60° for each turn. The results showed that the EIS curve remained consistent (Appendix A), and the corresponding Rct was around 0.47 Ω (Appendix A). The outstanding mechanical and electrochemical properties provide excellent application prospects for N-MXene-F-based SSC in flexible electronic-energy storage.

## 4. Conclusions

In summary, we devised an appropriate strategy to develop an MXene-based FSC with enhanced electrochemical performance. The MXene@MF composites were first prepared by electrostatic self-assembly between positively charged MF microspheres and negatively charged MXene nanosheets. Then, N-MXenes with 3D porous structures were obtained through an annealing process. Finally, we developed a free-standing, highly flexible N-MXene-F electrode, with excellent mechanical and electrochemical properties, combined with electrospinning. Benefitting from the abundant mesopores structure and uniform nitrogen atom doping, the symmetrical supercapacitors assembled from the N-MXene-F electrodes exhibit excellent electrochemical performance. Specifically, the specific capacitance at a current density of 1 A g^−1^ is 71 F g^−1^, and the areal capacitance is 54 F cm^−2^. An energy density of 12.78 Wh kg^−1^ was achieved at a power density of 1080 W kg^−1^. In addition, under different bending degrees, the electrochemical performance of N-MXene-F supercapacitors hardly changes, with excellent mechanical stability. Therefore, SSC devices based on ultra-flexible N-MXene-F have broad application prospects in flexible electronic energy-storage devices, especially next-generation wearable devices.

## Figures and Tables

**Figure 1 molecules-27-04890-f001:**
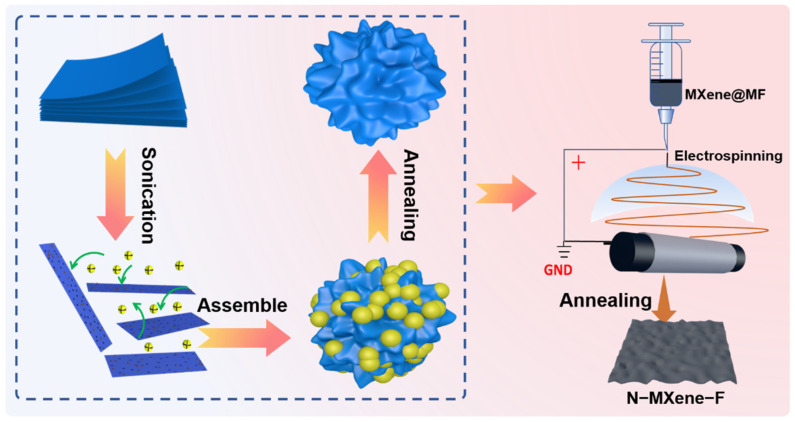
Schematic illustration of the fabrication of the N-MXene and N-MXene-F.

**Figure 2 molecules-27-04890-f002:**
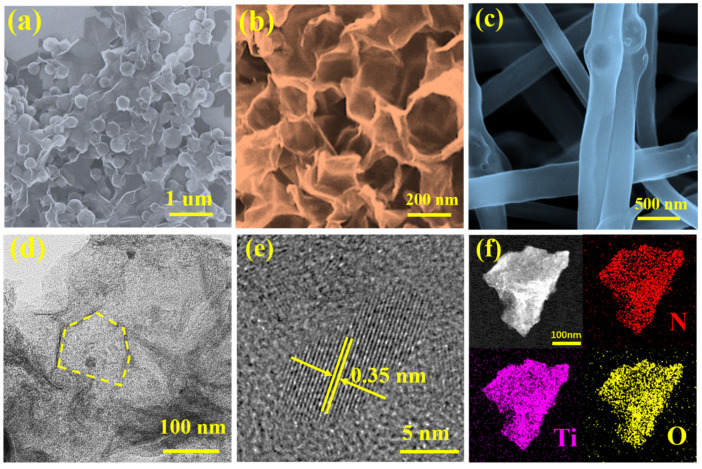
Morphology and structure characterizations of MXene@MF and N-MXene: SEM image of (**a**) MXene@MF; (**b**) N-MXene; (**c**) N-MXene-F; (**d**) TEM image of N-MXene; (**e**) HRTEM image of N-MXene; (**f**) STEM-EDX mapping of N-MXene.

**Figure 3 molecules-27-04890-f003:**
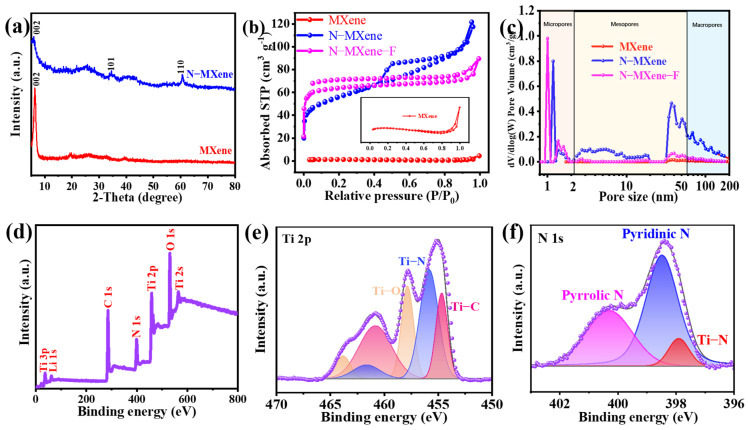
(**a**) XRD patterns of N-MXene and MXene; (**b**) nitrogen adsorption/desorption isotherms of MXene, N-MXene, and N-MXene-F; (**c**) pore size distribution of MXene, N-MXene, and N-MXene-F; (**d**) XPS survey scan; (**e**) Ti 2p; (**f**) N 1s spectra of N-MXene.

**Figure 4 molecules-27-04890-f004:**
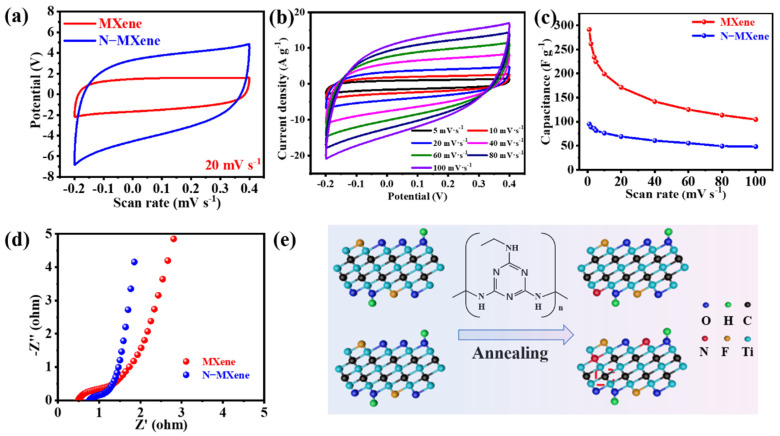
Electrochemical measurement by the three-electrode system: (**a**) CV curves of the N-MXene and MXene electrodes; (**b**) CV curves of the N-MXene at different scan rates ranging from 1 to 100 mV s^−1^; (**c**) specific capacitance of the N-MXene and MXene electrode at various scan rates; (**d**) Nyquist electrochemical impedance spectra; (**e**) schematic diagram of nitrogen atom doping.

**Figure 5 molecules-27-04890-f005:**
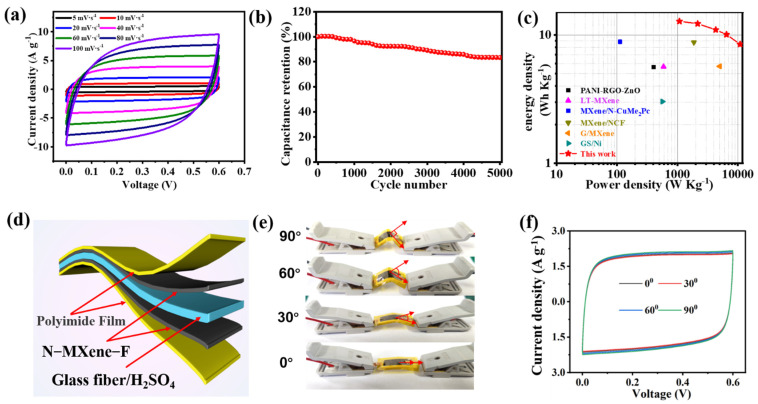
Electrochemical performance of the SSC: (**a**) CV curves at different scan rates; (**b**) specific capacitance retention of SSC at a current density of 10 A g^−1^; (**c**) Ragone plots displaying energy and power densities of N-MXene-F//N-MXene-F SSC in comparison with other MXene-based and graphene-based supercapacitors; (**d**) SSC assembly schematic; (**e**) digital photos of SSC with different bending angles; (**f**) CV curves of SSC under different bending angles.

## Data Availability

The data presented in this study are available on request from the corresponding author.

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
