# Peer review of "Nitrogen-Doped Porous MXene (Ti3C2) for Flexible Supercapacitors with Enhanced Storage Performance"

_molecules, 2022, doi:10.3390/molecules27154890_

Round 1

Reviewer 1 Report

The manuscript shows the fabrication of a flexible supercapacitor device based on MXenes infiltrated into flexible carbon fibers. The design and further development of flexible supercapacitor devices is important and relevant. The authors present an interesting approach to design such materials and the electrode shows a good performance as supercapacitor. However, a revision is needed to improve the quality of the work and to make it more sound for the community of supercapacitor researchers, in particular, when it comes to the mechanical properties of the device. 

1. P 1 line 35-40: The authors state that the challenge to design flexible devices is the construction of flexible devices. And as a result, new FSCs need to be developed. But why is this a challenge, the authors do not give any scientific fact about the challenges. There are many flexible materials such as polymers, fibers, even framework materials, which are flexible. The reason why it is difficult to apply them as a flexible device isn’t the material synthesis itself. It is the implementation and the materials properties that do not qualify them for application in supercapacitors. The authors should give real arguments such as the physical and chemical requirements for a material to be applied in flexible energy storage devices (conductivity, tensile strength, bending strength, surface area, and so on…). I am missing a logic argumentation in the introduction that leads to the outline of the paper.

2. P1 line 41-52: English needs major correction. There are a few typos, some sentences are missing words, some are not understandable.

3. Citation in the text is not consistent and at times confusing.

4. Line 57: Authors should change capacity to capacitance as they talk about supercapacitors.

5. Line 79: What do the authors mean by ‘rich porestructure’? That’s a very non-scientific term. Authors should rather state the pore volume, or pore width distribution.

6. Experimental Section: Specification of the used chemicals must be given. More experimental details about the characterization of materials is necessary. SEM: Were the samples coated with a conductive material, which sample holders were used, which detector was used. TEM: What was the accelerating voltage. XPS: Which source was used, how was the data processed (Shirley background? Referencing to C1s or something else? Was a charge compensation used? Which software was used for data processing). Nitrogen physisorption: How were the specific surface area determined (which p/p0 range), at which p/p0 was the pore volume taken from.

7. Line 141: Authors state that the diameter of each fiber is about 447 nm. However, Figure 2c indicates something else. There are thinner and thicker fibers as well. Authors should give a proper histogram (~100 fibers+) of fiber diameters as this is an important parameter for the fabrication of fiber felts, and diffusion investigations in electrochemical devices.

8. BET surface areas should be given without decimal numbers as the error of the method is larger than 0.5 m2/g.

9. A yield of the whole process is missing and should be given.

10.  Line 180. The charge storage mechanism is based on capacitance, not capacity. The energy storage, a capacitance, is related to the surface area that is accessible for the formation of a electrochemical double-layer. Authors must clarify this in the whole manuscript. They should not refer to active sites but rather to accessible area.

11. Line 211: 292 F/g for the capacitance of 1 electrode is not ‘ultra-high’. Authors should try to circumvent such exaggerations. Some N-doped carbon materials with high surface area achieve such capacitance as well. Authors should rather give comparisons with other N-doped MXene materials in terms of capacitance and rather refer to Figure 4c for such values.

12. Instead of schematic Figures (Figure 5), the authors should show actual photographs of the device in a straight and bended configuration.

13. In terms of stability and mechanical properties of the device I suggest the following experiment: Bend the device to 90° and then bend it back stepwise and measure the impedance and the CVs again for 60°, and 30° and straight. The reversibility of the bending is as important as the first bending and in particular for wearable devices extremely important. Authors should also add the following experiment: Bend the device to 90°, bend it back to 0°, repeat for 10-20 times and report the capacitance retention. This is more important for flexible devices than the usual long-cycle stability test.

Reviewer 2 Report

This paper reports the development of flexible supercapacitors using nitrogen-doped porous MXene as the active material. The composition and morphology of the prepared N-doped porous MXene material have been characterized by the authors using several different techniques, including XRD, SEM, TEM, EDS, BET surface area and XPS. The assembled flexible supercapacitors exhibited high energy density and reasonably good stability after 5000 cycles. Overall, this work has enough merit to be published in Molecules journal. However, some revisions are needed before it can be accepted as detailed below:

1. The authors have provided the BET surface area data. However, the corresponding pore size distribution data of MXene and N-MXene  are missing. Please provide them.

2. The reasons behind the significant increase in the BET surface area of N-MXene  compared to pristine MXene should be provided.

3. The morphology of N-MXene after cycling can be checked to determine the structural stability. So, please check it by SEM.

4. How much are the N, C, Ti, and O contents in MXene (Figure 4c-f) based on the XPS analysis?

5. The peaks in Figure 3a can be labelled more properly.

6. In the Experimental Section, the authors can provide one more section for the chemicals and they need to provide a list of all chemicals used in this work and the purity of these chemicals.

7. In Section 2.2, the heating rates used for calcination processes should be specified.

8. The term "electrostatic spinning" should be changed to "electrospinning" which is a more commonly used term.

9. In the Introduction, more recent references on the development of different types of electrode materials for supercapacitors, such as Chemical Communications 58 (7), 1009-1012; ACS Appl. Mater. Interfaces, 12, 39154–39162 (2020); Nano Convergence 9 (1), 1-12 (2022);  Chem. Eng. J., 385, 123454 (2020);  Nano Energy 52, 336-344 (2018) and Electrochimica Acta, 394, 139058 (2021) can be mentioned and cited.
